# In Silico Mass Spectrometric Fragmentation and Liquid Chromatography with Tandem Mass Spectrometry (LC-MS/MS) Betalainic Fingerprinting: Identification of Betalains in Red Pitaya

**DOI:** 10.3390/molecules29225485

**Published:** 2024-11-20

**Authors:** Jesús Alfredo Araujo-León, Ivonne Sánchez-del Pino, Ligia Guadalupe Brito-Argáez, Sergio R. Peraza-Sánchez, Rolffy Ortiz-Andrade, Victor Aguilar-Hernández

**Affiliations:** 1Unidad de Biología Integrativa, Centro de Investigación Científica de Yucatán, A.C., Mérida 97205, Yucatán, Mexico; jalfredoaraujo@gmail.com (J.A.A.-L.); lbrito@cicy.mx (L.G.B.-A.); 2Unidad de Recursos Naturales, Centro de Investigación Científica de Yucatán, A.C., Mérida 97205, Yucatán, Mexico; isanchez@cicy.mx; 3Unidad de Biotecnología, Centro de Investigación Científica de Yucatán, A.C., Mérida 97205, Yucatán, Mexico; speraza@cicy.mx; 4Facultad de Química, Universidad Autónoma de Yucatán, Mérida 97069, Yucatán, Mexico; rolffy@correo.uady.mx

**Keywords:** betacyanins, betacyanin derivatives, betaxanthins, LC-MS/MS, fingerprint betalains, red pitaya

## Abstract

Betalains, which contain nitrogen and are water soluble, are the pigments responsible for many traits of plants and biological activities in different organisms that do not produce them. To better annotate and identify betalains using a spectral library and fingerprint, a database catalog of 140 known betalains (112 betacyanins and 28 betaxanthins) was made in this work to simplify betalain identification in mass spectrometry analysis. Fragmented peaks obtained using MassFrontier, along with chemical structures and protonated precursor ions for each betalain, were added to the database. Product ions made in MS/MS and multistage MS analyses of betanin, beetroot extract, and red pitaya extract revealed the fingerprint of betalains, distinctive ions of betacyanin, betacyanin derivatives such as decarboxylated and dehydrogenated betacyanins, and betaxanthins. A distinctive ion with *m*/*z* 211.07 was found in betaxanthins. By using the fingerprint of betalains in the analysis of red pitaya extracts, the catalog of betalains in red pitaya was expanded to 86 (31 betacyanins, 36 betacyanin derivatives, and 19 betaxanthins). Four unknown betalains were detected to have the fingerprint of betalains, but further research will aid in revealing the complete structure. Taken together, we envisage that the further use of the fingerprint of betalains will increase the annotation coverage of identified molecules in studies related to revealing the biological function of betalains or making technologies based on these natural colorants.

## 1. Introduction

Betalains are water-soluble natural pigments that contain nitrogen and have interesting absorption and emission properties; they are backed by conjugated double bonds and at least one heterocyclic nitrogen. Studies on the properties of betalains have revealed that, besides their spectroscopic and fluorescent properties, betalains are essential for attracting pollinators and dispersers by providing color to plant tissues; contribute to drought tolerance and excessive light stress [1]; and are important for preserving good health through their diverse biological activities, such as anti-inflammatory, antioxidant, anti-hypertensive, antidiabetic, and immunomodulatory activities, as well as through their cancer chemoprotective properties [2,3]. Another health benefit of betalains has been seen when betalains are used as a natural food colorant, which has been approved by the Food and Drug Administration (FDA) and by the National Food Safety Standard in China (National Food Safety Standard Use for Food Additive, GB2760-2011) [4]. However, the different biological activity tests and the color of the plant tissues correlate with the betalains contained in the extracts. The assertion that specific betalains are present points out the value of diverse sources of betalains, suggesting that analytical methods are crucial for accurately identifying betalains with high sensitivity. Sources of betalains include most families of plants in the order Caryophyllales, fungi of the genus *Amanita*, and the bacteria *Gluconacetobacter diazotrophicus* [5,6,7]. Biochemical, genetic, and evolutionary studies have revealed that the lineage-specific production of betalains in the order Caryophyllales lies in the conservation and duplication events of genes encoding enzymes in the biosynthetic pathway of betalains [8].

Two main classes of betalains exist: red-violet betacyanins and yellow-orange betaxanthins. In plants, during the biosynthesis of betalains, betalamic acid is condensed with either *cyclo*-DOPA or glycosylated *cyclo*-DOPA. The sugar moieties can be acylated with aromatic organic acids, such as ferulic, *p*-coumaric, caffeic, sinapic, and malonic acid, to further prevent the cleavage of betalains by β-glucosidases. Betaxanthins result from condensed betalamic acid with an imino or amino group of amino acids. Extra expansion of the chemical diversity of betalains is due to epimerization at the chiral center C15 [9]. By contrast, the degradation of betalains involves decarboxylation by removing carboxyl groups and oxidative dehydrogenation. More factors that impact the stability of betalains have been noted to result in damaging effects on the pigment. These causes include the temperature of food storage, pH, and light, among other causes, which are extensively reviewed in Sadowska-Bartosz and Bartosz 2021 [10]. Plant edibles of the order Caryophyllales that produce betalains include roots of *Beta vulgaris* (beetroot), tubers such as *Ullucus tuberosus* Caldas (Ullucos), leaf of *B. vulgaris L.* ssp. Cicla (Chard), grain and vegetable amaranth classes (*Amaranthus cruentus* L., *A. caudatus* L. and *A. hybridus* L., *Amaranthus tricolor*), and fruits of the genus *Opuntia*, *Eulychnia*, and *Selenicereus (Hylocereus)* [10].

Fruit of the genus *Selenicereus* exist in three different colors assigned to betalains [11,12]: red-skinned fruit with white flesh (*Selenicereus undatus*), red-skinned fruit with red flesh (*S. costaricensis* and *S. monacanthus*), and pink-skinned fruit with white flesh or yellow-skinned fruit with white flesh (*S. megalanthus*). Among those species, *S. undatus* is the most widely spread and cultivated around the world, including countries such as Vietnam, China, Indonesia, the United States, and Mexico, among others. In *Selenicereus* spp., as in other Caryophyllales plants, the lack of analytical standards for characterizing betalains has been managed by the chemical synthesis of betalains and purifying and separating individual endogenous betalains to reveal the biological activities supported by them [13,14,15,16,17]. However, key biological activities have been revealed for betalains in *Selenicereus* spp., such as antioxidant activity [18,19,20]; the thermal stability of betacyanins from juice [21,22,23,24]; the stability of betalains in pure form, in spray-dried pitaya peel powders, or as a natural colorant in food [25,26,27]; antibacterial activity [28,29,30]; and antiviral activity against IAV (influenza A virus)-infected A549 cells [31]. As multiple betalains may exist, as in other Caryophyllales plants, identifying betalains needs extensive coverage to assess the influence of the diverse array of betalains.

Structural characterization and massive surveying of betalains through either Nuclear Magnetic Resonance (NMR) or mass spectrometry (MS) not only provides opportunities to understand the biology of betalains but also helps to distinguish unrevealed betalains. For example, the preparative fractionation of betalains from *Phytolacca americana* has resulted in the discovery of structural features of *15S*-betanin and *15R*-isobetanin and an expansion of the betalains catalog to 17 betalains [16,32]. The catalog of betalains has been further expanded by liquid chromatography (LC) coupled with MS and the fortunate discovery of distinct retention times of betalain epimers during chromatography analysis, which typically display different retention times. Although candidates for betalains were identified at all five confidence levels [33,34], most of them were identified at level 4 with molecular formulas by means of charge state determination, adduct ion determination, isotope abundance distribution, and UV–Vis absorption [35,36,37]. The catalog of betalains has been expanded to 22 betalains (15 betaxanthins and 7 betacyanins) in beetroot [38], 24 betalains (18 betaxanthins and 6 betacyanins) in 10 Mexican prickly pear cultivars [39], 28 betalains (19 betaxanthins and 9 betacyanins) in Swiss chard [40], 43 betalains (30 betaxanthins and 13 betacyanins) in *Amaranthus cruentus* [37], 48 betalains (22 betaxanthins and 26 betacyanins) in three beetroot cultivars [41], 68 betalains (34 betaxanthins and 34 betacyanins) in Djulis [36], and 146 betacyanins, most of which have a high molecular weight of over 1000 Da, in *Bougainvillea glabra* [42]. Given that some betacyanins are specific to certain species, the combined number of betacyanins from different species exceeds 200 [42,43], which highlights the wide variety of these pigments.

This work aims to examine the fingerprint of betalains in samples that are compatible with mass spectrometry by means of MSPD (Matrix Solid-Phase Dispersion)-based preparation samples, LC-MS/MS analysis, and spectra interpretation aided by in silico spectral libraries of betalains.

## 2. Results and Discussion

### 2.1. In Silico Fragmentation Library of Betalains Using MassFrontier Software

To better understand the natural diversity of betalains, we exploited spectral libraries to achieve structural characterization, where the fragment peaks and precursor mass work together to reveal the features of each pigment. We began by conducting a survey of all known betalains [15,36,42,43], then created chemical structures and precursor masses for in silico fragmentation using MassFrontier software 7.0 [44]. In the database, there were 140 betalains. Betalains can then be grouped into betacyanins and betaxanthins (Figure 1). The database has more betacyanins than betaxanthins, with 112 betacyanins and 28 betaxanthins, which correspond to 80% and 20%, respectively. Then, when the betacyanins were classified into betacyanin classes as reported by Xie et al., 2021 and Kumorkiewicz-Jamro et al., 2021 [36,43], the dominant betacyanin class was betanin-type, with 39.4% of the betacyanins, followed by amaranthin-type, melocactin-type, apiocactin-type, gomphrenin-type, and glabranin-type. When betaxanthin classes were reported by Esteves et al., 2022 [15], the dominant class was the hydrophobic-type with 39.3%, followed by special cases that include r-aminobutyric acid-bx(betaxanthin), dopamine-bx, L-DOPA-bx, tyramine-bx, 3-methoxy-tyramine-bx, 5-hydroxynorvaline-bx, methionine sulfoxide-bx, the positively charged-type, the polar uncharged-type, and the negatively charged-type.

When the molecular mass distribution of betalains was analyzed, the molecular ion [M+H]^+^ masses of betalains ranged from 250 to 1350. However, the molecular ion [M+H]^+^ masses of betacyanins and betaxanthins do not overlap (Figure 2). Betaxanthins with smaller molecular ion [M+H]^+^ masses map up to 400, while betacyanins not only have larger molecular masses but also display isomeric forms at C-15 (Figure 3A). For instance, betanin and isobetanin, or the aglycone of almost all betacyanins known as betanidin, with its isomeric form being isobetanidin. Betacyanin’s molecular ion [M+H]^+^ masses being larger than betaxanthins is explained by the attached glycosyl groups, while betaxanthins contain an amino group of amino acids. The database displayed in Appendix A also shows the chemical structure, chemical formula, monoisotopic mass, [M+H]^+^ ion, and the product ion mass list of [M+H]^+^ for each betalain. This library compiles a list of both shared and specific ion mass lists for betacyanins and betaxanthins. They share the molecular frame of betalamic acid, and the structure of aglycone has the frame of all betacyanins, with *m/z* 389.0979, except for decarboxylated or dehydrogenated betacyanins. The specific framework for betacyanin-types are as follows: betanin-type with *m/z* 551.1508, amaranthin-type with *m/z* 727.1829, melocactin-type with *m/z* 713.2036, apiocactin-type with *m/z* 683.1930, gomphrenin-type with *m/z* 551.1508, glabranin-type with *m/z* 713.2036, and specific ions for betaxanthin classes. In betacyanins, extra shifts of *m/z* are associated with the addition of substituents or the loss of an H_2_ or carboxylic group. For instance, Figure 2 shows the 16 amaranthin-type betacyanins, which are heavier and lighter than amaranthin. Currently, this database is both new and has the largest collection of betalains available, as far as we know.

### 2.2. Fingerprint Mass Spectrometry of Betalains Using Betanin and Betanidin

To highlight and support the main fragments in the made-in-silico fragmentation library of betalains that can aid in the definition of the betalain fingerprint, we used a previously reported untargeted metabolomics platform [37] that identified the betanin and its isomer in a beetroot extract from a SIGMA vendor (Cat. 901266; Figure 3). Betanin differs from isobetanin at the chiral center C15, and their retention times were 3.46 and 6.25 min, as displayed in an extract ion current chromatogram (XIC) with *m/z* 551.1495 (Figure 3B). The chromatographic resolution of enantiomers in betalains is not surprising; it did not require the use of columns with a chiral stationary phase. For instance, longer retention for isobetanin in C18-based reversed-phase chromatography than betanin has been reported in numerous studies [36,37,45,46]. The difference in chromatographic retention time between betanin and isobetanin has been attributed to their unique stereochemistry at the C-15 position, which affects their interaction with the chromatographic column and, thus, their elution times [9]. Taking advantage of the link between structural features and fragments of betanin in the in silico library, the peak with *m/z* 389.09 in the betanin spectrum was annotated as the aglycone of betanin, also known as betanidin/isobetanidin.

Next, to explore the fragmentation of the betanidin in more detail, which is the aglycone of almost all betacyanins with *m/z* 389.09, our acquisition of MS/MS via fragmentation with high-energy HCD at 50 eV in a data-dependent acquisition experiment was compared with an analysis in multistage MS with collision-induced dissociation (CID) (Figure 4). Key fragmentation features were discovered in the analysis, including common product ions such as *m/z* 345.10 due to the loss of a carboxyl group and *m/z* 343.09 due to the loss of a carboxyl group plus H_2_, and the ions with *m/z* 150.05, 194.04, 178.05, 132.04, and 106.06. Decarboxylated betanidin followed 5,6-dihydroxy-indole, and two carbonyls attached to the pyridine ring were prominent feature assignments. Further, to reveal other ions of betalains for the straightforward identification of betalains, the betanin MS^2^ spectra were surveyed for ions with relative abundance higher than 5% and matching structure (Appendix A). Up to 23 ions with *m/z* between 389.09 and 106.06 were discovered. Thirteen ions with *m/z* greater or equal to 253.09 and ions with *m/z* 176.07 and 150.05 were matched with exclusive structural sections of betacyanins, whereas eight distinct ions with *m/z* 211.07, 194.04, 179.08, 178.05, 166.05, 138.05, 132.04, and 106.06 were matched with the core structure of betalains containing the pyridine ring. The structural information deduced for the ions found here suggests that they are essential fragments for identifying betalains.

### 2.3. Identification of Betalains in Beetroot Extract

We used a previously reported method compatible with mass spectrometry sample preparation based on Matrix Solid-Phase Dispersion (MSPD) [37] that extracted betalains from beetroot extract. The analysis of homemade beetroot extract was chosen as it would survey a large population of betalains. The mass spectrometry data, derived from data-dependent acquisition with high-energy HCD fragmentation, was interrogated by XIC, selecting features harboring the core ions of betalains as product ions, including the ions with *m/z* 211.07, 194.04, 179.08, 178.05, 166.05, 138.05, 132.04, and 106.06. A total of 36 betalains were identified (Appendix A). Furthermore, 26 betacyanins in the collected betalains were identified using the ions with *m/z* 345.10, 343.09, 299.10, 297.08, 281.09, 269.09, 255.11, 253.09, and 389.09, corresponding to the aglycone of almost all betacyanins, except for betacyanin derivatives that are decarboxylated and dehydrogenated betacyanins. The ions with *m/z* 389.09 and 150.05 have been seen in the analysis of betalains from *Chenopodium formosanum* and *Amaranthus cruentus* [36,37].

The betacyanin derivatives do not produce a product ion with *m/z* 389.09 because the aglycone frame has lost at least one carboxylic group or has been dehydrogenated. The identification of betacyanin derivatives was markedly straightforward with the fingerprint ions of betalains, but the confirmation of the carbons undergoing decarboxylation will need further investigation since mass spectrometry is not able to specify carbons undergoing decarboxylation in these conditions. Based on the data for decarboxylated betacyanins, for which the decarboxylated betacyanins at either C2, C15, or C17 were resolved in C18-phase chromatography [32,33,34], the decarboxylated betacyanins identified here were mapped relative to betanin. Betacyanin derivatives in the beetroot extract were composed of nine betanin derivatives and four neobetanin derivatives. Decarboxylated betacyanin derivatives displayed the loss of a carboxylic group in the framework; thus, the precursor ion mass was smaller than their counterpart, unmodified betacyanin. For instance, the mass of the betanin precursor ion was *m/z* 551.1483, and the mass of the 17-decarboxy-betanin precursor ion was *m/z* 507.1594. Decarboxylated betacyanin derivatives showed the typical loss of a glucosyl moiety with an *m/z* of 162.053 [M + H −162.053], which explains, for instance, the product ion with *m/z* 345.10 or *m/z* 343.09 in decarboxylated betanins (2-decarboxy-betanin, 15-decarboxy-betanin, and 17-decarboxy-betanin) and decarboxylated neobetanin (for example, 2-decarboxy-neobetanin), respectively (Figure 5). Detecting those ions was in accordance with previous reports of betalain analysis [22,36,47,48]. As decarboxylated betacyanin derivatives, dehydrogenated betacyanin derivatives showed a loss of H_2_ in their framework; thus, the precursor ion mass was smaller than the counterpart unmodified betacyanin, as previously described [49]. Dehydrogenated betacyanins and betacyanin derivatives include neobetanin with *m/z* 549.13, 2-decarboxy-2,3-dehydro-betanin with *m/z* 505.1435, 2-decarboxy-2,3-dehydro-isobetanin with *m/z* 505.1429, 2,17-bidecarboxy-2,3-dehydro-betanin with *m/z* 461.1538, 2-Decarboxy-2,3-dehydro-neobetanin with *m/z* 503.1281, and 2,17-Bidecarboxy-2,3-dehydro-neobetanin with *m/z* 459.134.

Next, betaxanthins were analyzed based on the hypothesis that a fingerprint of betalains can be generated upon molecular ion fragmentation in a mass spectrometer since betacyanins and betaxanthins share the framework of betalamic acid. Indeed, 13 betaxanthins were identified in the beetroot extract, including 10 hydrophobic-type betaxanthins and 3 polar uncharged-type betaxanthins. Remarkably, a product ion with *m/z* 211.07 was found in betaxanthins that structurally maps to betalamic acid, bearing a nitrogen attached to the corresponding amino acid of betaxanthins. Additionally, ions were structurally mapped to the heterocyclic pyrimidine ring, such as those with *m/z* 194.04, 166.05, 138.05, 132.04, 130.05, and 106.06. For instance, the generation of these ions was found in the mass spectra of glutamine-bx and tryptophan-bx (Figure 6). Given the variety of conjugated amino acids in betaxanthins, defining distinct ions with *m/z* values that match to a framework outside of the ion with *m/z* 211.07 was a challenge.

### 2.4. Identification of Betalains in Red Pitaya Extract

Assuming that our identified betalain features in beetroot, along with the betalain fingerprint ions determined in this study, can be used to survey betalains in red pitaya, we examined the red pitaya extract made following the previously reported method based on MSPD that is compatible with mass spectrometry [37]. A total of 86 betalains were identified in the red pitaya extract distributed at 12 min chromatographic resolution with different abundance (Table 1 and Figure 7). The betalains discovered in this study were previously identified by others using chemical, LC-MS/MS, or NMR methods [15,36,37,43,50,51,52], with the exception of four unknown betalains. The abundance of each betalain was determined by measuring the area below the base peak of the ion *m/z* of betalain that was examined. When the betalains in the extract were classified into two main classes, it was notable that all 19 identified betaxanthins were of low abundance compared to the total 67 betacyanins that map to *m/z* above 450. This work expanded the catalog of betalains in pitaya to 86 compared with the previous number of betalains reported. For instance, 8 betalains in red-purple pitaya *Hylocereus polyrhizus* (Weber) Britton & Rose [53]; 11 in the red flesh and orange flesh varieties of *Stenocereus pruinosus* and *Stenocereus stellatus* [54]; 20 in the red flesh of *Hylocereus costaricensis*, white flesh of *Hylocereus megalanthus,* and *Hylocereus undatus* [55]; and 23 in the red flesh of *Hylocereus polyrhizus* and *Hylocereus undatus* [56].

### 2.5. Annotating Unknowns with the Fingerprint of Betalains

Surprisingly, we saw two unknown betalains in red pitaya during the analysis of mass spectrometry data made in this study. Those betalains, referred to here as unknown-1 and unknown-2, were not contained in our database or previously reported, as far as we know. They were characterized by the molecular formula C_35_H_42_N_2_O_21_ and by MS/MS spectra containing the distinctive ions of betalain with *m/z* 389.09, 345.10, 343.09, 297.08, 194.04, 150.05, and 106.06. Unknown-1 had an *m/z* of 827.2316 and eluted at 8.83 min, while unknown-2 had an *m/z* of 827.2328 and eluted at 9.23 min. Given that betalains have isomeric forms, we suspected that unknown-1 was the isomeric form of unknown-2. Next, we decided to use multistage MS with CID to reveal the identity of neutral losses in the unknown betalains and compare it to the MS/MS with HCD fragmentation (Figure 8). The spectrum obtained with HCD displayed the typical ions with *m/z* 389.09, 194.04, 150.05, and 106.06 of betalains. Next, fragmenting the parent ion with *m/z* 827.2316 using CID revealed the neutral loss of 134.0423 Da, corresponding to a pentoside or glutaric acid. Then, fragmenting the product ion with *m/z* 695.1930 using CID revealed a neutral loss of 162.0528 Da, matching to a glucoside. Also, fragmenting the product ion with *m/z* 551.1508 revealed a neutral loss of 162.0528 Da, corresponding to a glucoside. Although these data reveal the decorating groups on the framework of betalain, further research will aid in revealing the complete structure of unknown-1 and unknown-2. Interestingly, both unknowns had decarboxylated forms in the extract that eluted at 10.27 and 10.54 min with an *m/z* of 783.2455.

## 3. Materials and Methods

### 3.1. Chemicals and Reagents

Analytical-grade methanol (MeOH) and water (H_2_O) were purchased from J.T. Baker (Phillipsburg, NJ, USA). Analytical-grade formic acid and acetic acid were obtained from Merck (Sigma Aldrich, St. Louis, MI, USA). As an analytical standard, a betanin red beet extract diluted with dextrin was used (Sigma Aldrich, St. Louis, MI, USA). Methanol and water (mass spectrometry-MS grade) were purchased from TEDIA (Fairfield, OH, USA). Ultrapure Helium gas (>99.999%) was obtained from a local supplier (INFRA, Merida, Mexico). N2 was generated in-house using an NM32LA PEAK Generator (PEAK Scientific System, Inchinnan, Scotland, UK).

### 3.2. Fruit Juice Extraction and Matrix Solid-Phase Dispersion Extraction (MSPD)

Dragon fruit (*Hylocereus costaricensis*) juice was obtained using an electronic mill. Subsequently, the tissue debris was separated from the juice and then discharged. After collection, the juice was frozen and kept at −80 °C until it was lyophilized for 48 h (chamber temperature −50 °C and vacuum pressure −460 mmHg). The dry material was kept in darkness at −20 °C until it was used.

Total betalains were extracted from 100 mg of dry material using the Matrix Solid-Phase Dispersion (MSPD) method as described by Araujo-Leon et al., 2023 [37]. A homogenous mix of 400 mg of BondElut-C18 (Agilent Technologies, San Jose, CA, USA) and 100 mg of dry material was packed in an empty SPE cartridge with frits to minimize diffusion. Bound betalains were eluted with 9 mL of H_2_O with 0.1% acetic acid and then with 9 mL of MeOH:H_2_O (1:1, *v/v*) with 0.1% acetic acid using a vacuum manifold system (Visiprep SPE Vacuum Manifold, Sigma-Aldrich, Burlington, MA, USA). Separately, the eluents were collected and evaporated. The residues were resuspended in 0.6 mL of H_2_O with 0.1% acetic acid and transferred to an amber vial (MS-grade) for LC-MS analyses.

### 3.3. UHPLC-UV-VIS-MS/MS Orbitrap

The red pitaya extracts were analyzed by means of a UHPLC Ultimate 3000 system (Thermo Scientific, Waltham, MA, USA) equipped with a quaternary solvent manager, autosampler, column heater, and UV–Vis detector and coupled to a LTQ-Orbitrap Elite mass spectrometer (Thermo Fisher Scientific, San Jose, CA, USA) equipped with a Heated Electrospray Ionization interface (HESI-II, Thermo Fisher Scientific, Waltham, MA, USA). The mobile phase consisted of H_2_O with 0.1% acetic acid (solvent A) and MeOH with 0.1% acetic acid (solvent B) delivered according to the following gradient profile: 95% A kept for 2 min; 5% B to 100% B in 18 min; 100% B to 5% and isocratic for 10 min for column reconditioning. The sample was eluted from the column at a flow rate of 300 μL/min. The red pitaya extracts (20 μL) were injected into the UHPLC system equipped with a Hypersil GOLD C18 (Thermo Scientific, Waltham, MA, USA) reversed-phase column with 100 mm length, 2.1 mm internal diameter, and 1.9 μm particle size. The chromatographic column was kept at 45 °C ± 0.5 °C. UV/Vis spectra data were collected at 480 and 540 nm.

The electrospray ionization was optimized as follows: spray voltage, +4.5 kV; capillary temperature, 300 °C; heater temperature, 180 °C; sheath gas flow rate, 40; auxiliary gas, 15. MS spectra data were acquired in positive mode. Multidimensional MS (MS^n^) data were acquired to support the proposed structure hypothesis. For data acquisition under data-dependent acquisition (DDA) mode: MS1 resolution 60,000, scan range 100–1500 *m/z*; and MS2 resolution 60,000, scan range 100–600 *m/z*. Higher-energy collisional dissociation (HCD) was used as a fragmentation method by applying 50 eV. N_2_ and helium (He) were applied as cone and collision gases, respectively. Peak areas of each compound based in MS1 mode using the extracted ion chromatograms were determined with Xcalibur 4.1 software (Thermo Scientific, Waltham, MA, USA). Graphs were obtained using GraphPad Prism version 9.0 (GraphPad, San Diego, CA, USA).

### 3.4. MassFrontier-Based In Silico Fragmentation Library

Chemical structures were drawn for known betalains [15,36,43] using ChemDraw Professional 16.0 in accordance with the guidelines of the American Chemical Society (ACS-1996). Next, the structures were placed in the Structure Editor of MassFrontier 7.0 software (Thermo Scientific, Waltham, MA, USA). The charge specification was made to generate precursor masses [M+H]^+^ and fragment ions in silico. HighChem ESI Pos 2008 and HighChem Fragmentation Libraries were used. The fragmentation parameters were established to be steric optimal, allowing resonance reactions for electron sharing, charge stabilization, and radical isomerization. Fragment peaks and precursor masses were deposited in Appendix A.

## 4. Conclusions

Collectively, our compiled database of betalains, as provided in the Appendix A of this manuscript, and the mass spectrometry fingerprints of betanin, beetroot extract, and red pitaya extract revealed information about betalain fragmentation. These data can aid in interpreting the mass spectra of betalains, which ultimately not only improves the identification of betalains but also serves as a guide for mass spectra annotation. The aglycone betacyanin-derived ions with *m/z* 389 for betacyanins, *m/z* 345 for betalain derivatives, and *m/z* 343 for neobetanin variants can be used for the rapid identification of decarboxylated and dehydrogenated betalains; further automation on parallel metabolomics studies will aid in increasing the annotation quality of metabolite identification. The ion with *m/z* 211.07 was characteristic of betaxanthins. As betalains are responsible for biological traits in Caryophyllales plants or for nutritional and biological activities seen in organisms feeding on raw or processed food containing betalains, further investigation will benefit from the improved identification of betalains. For red pitaya betalains, given the catalog has grown to 86, it is crucial to conduct more research into the diversity and abundance of betalains in different pitaya varieties with distinct flesh and skin colors.

## Figures and Tables

**Figure 1 molecules-29-05485-f001:**
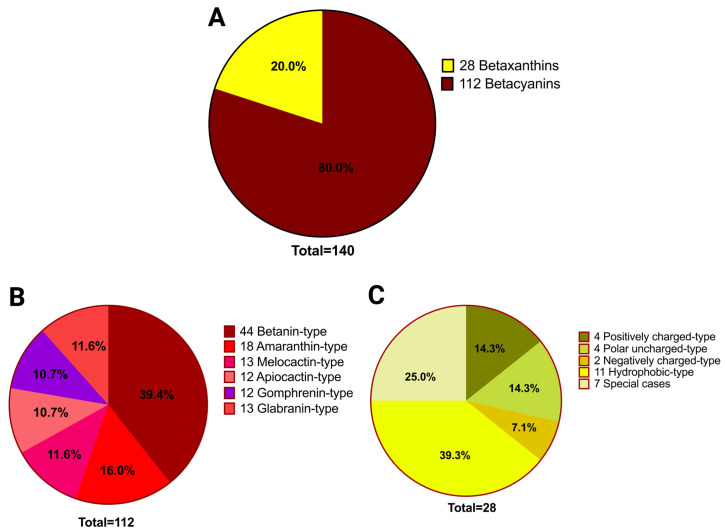
An outline of the classes of betalains used in the spectral library. (**A**) Betalain-types; (**B**) betacyanin-types; and (**C**) betaxanthin-types.

**Figure 2 molecules-29-05485-f002:**
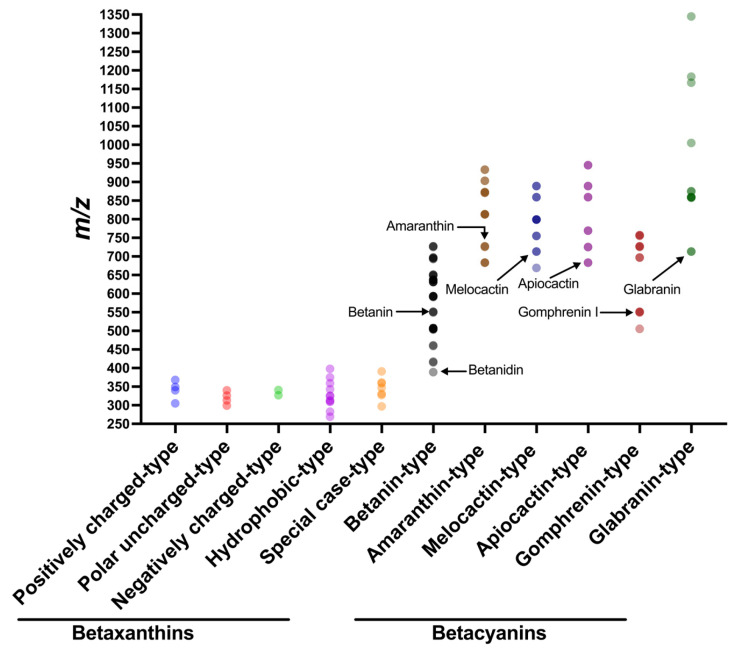
Distribution of the parent ion masses of betaxanthin and betacyanin classes. There are 140 betalains displayed, but many of them are isomeric.

**Figure 3 molecules-29-05485-f003:**
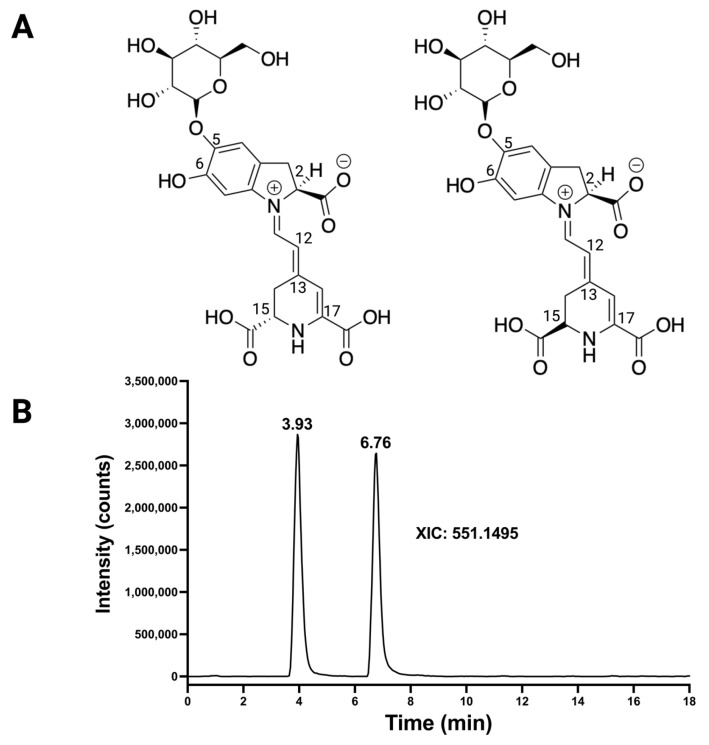
Identification of betanin and its isomer isobetanin. (**A**) The chemical structure of betanin and isobetanin. (**B**) Extract ion current chromatogram obtained for the ion *m/z* 551.1495.

**Figure 4 molecules-29-05485-f004:**
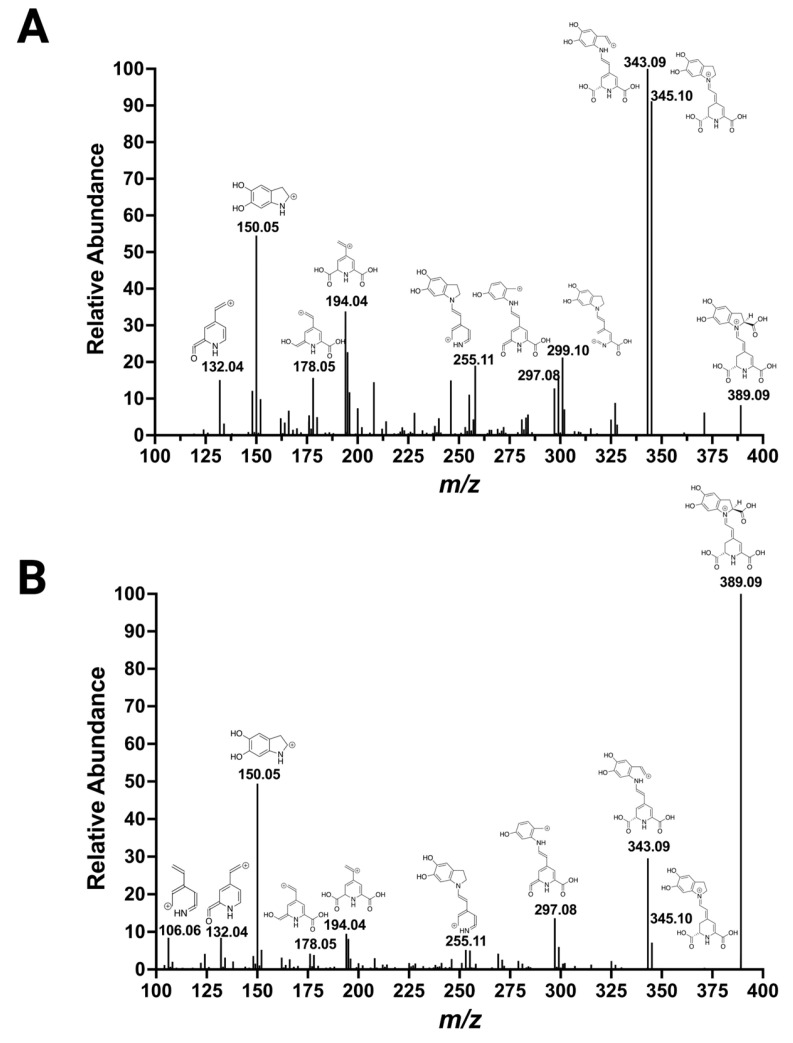
Fragmentation spectrum of betanin. (**A**) MS/MS spectrum of the ion *m/z* 551.1495 fragmented with high-energy HCD at 50 eV. (**B**) MS^3^ spectrum obtained with the ion with *m/z* 389.09 fragmented with CID at 35 eV (551.1495 CID35 → 389 CID35 →).

**Figure 5 molecules-29-05485-f005:**
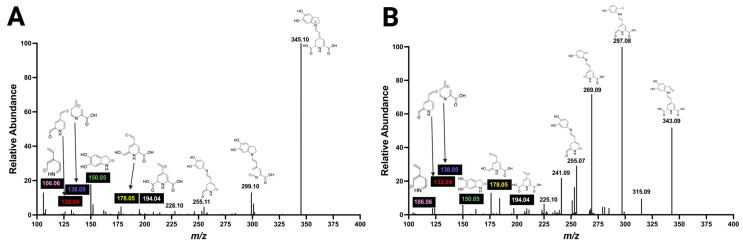
Fragmentation spectra of betacyanin derivatives with HCD at 50 eV. (**A**) MS/MS spectrum of 2-decarboxy-betanin. (**B**) MS/MS spectrum of 2-decarboxy-neobetanin. Core ions are highlighted in black.

**Figure 6 molecules-29-05485-f006:**
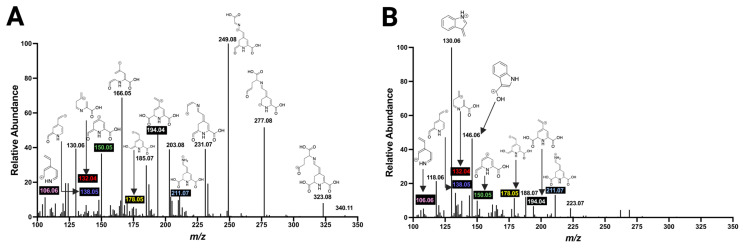
Fragmentation spectra of two betaxanthins with HCD at 50 eV. (**A**) MS/MS spectrum of glutamine-bx. (**B**) MS/MS spectrum of tryptophan-bx. Core ions are highlighted in black.

**Figure 7 molecules-29-05485-f007:**
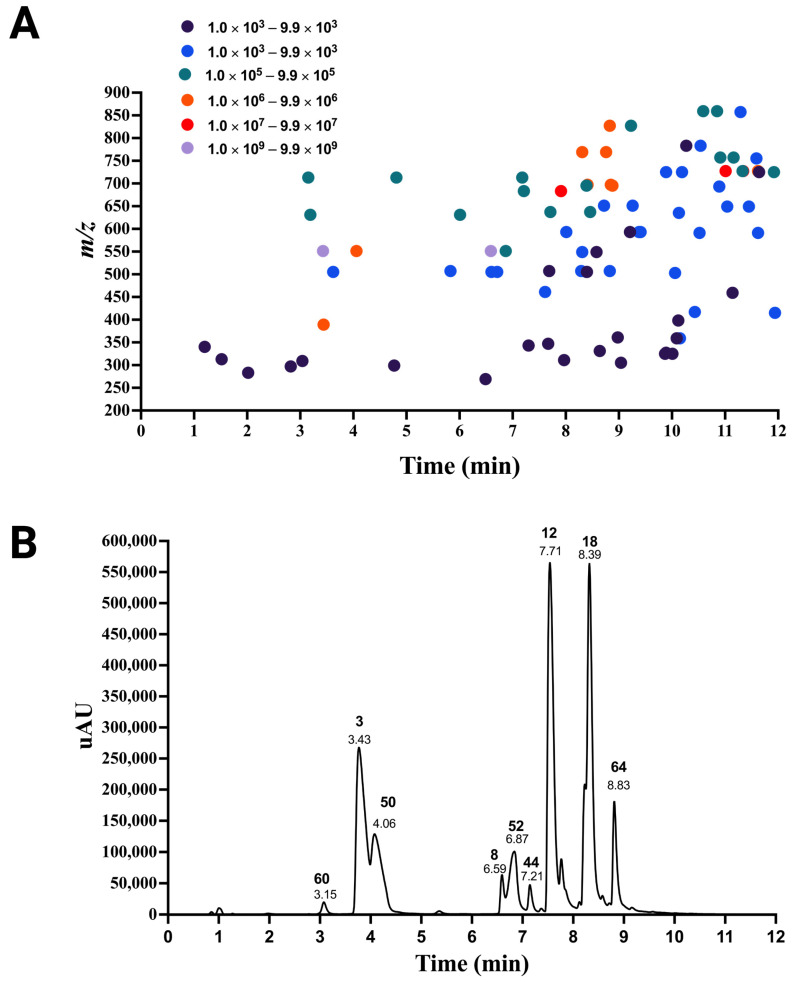
Resolving the betalains discovered in red pitaya. (**A**) The color circle scale indicates the change in the parent ion base peak measurement area for betalains. (**B**) Chromatogram of red pitaya at 540 nm. The peak displays the retention time and label number of betalain. Table 1 contains the complete list of identified betalains.

**Figure 8 molecules-29-05485-f008:**
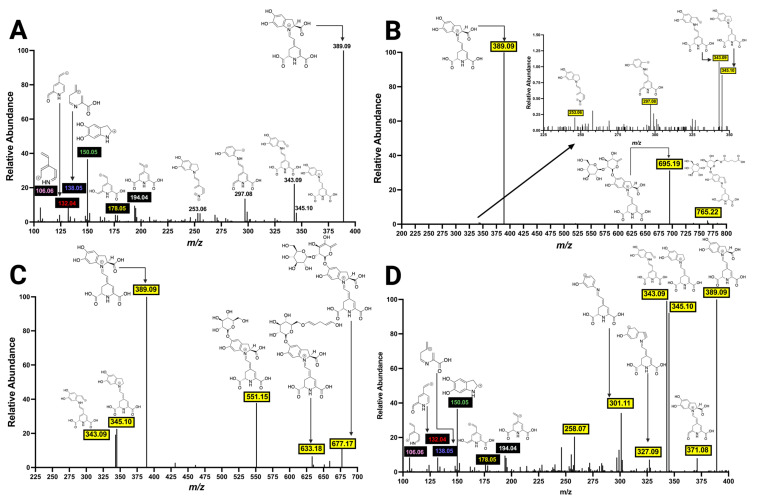
Spectra of unknown-2 betacyanin. (**A**) MS/MS spectrum obtained with fragmentation of ion *m/z* 827.23 with HCD. (**B**) MS/MS spectrum obtained with the ion *m/z* 827.23 fragmented with CID at 35 eV. (**C**) MS^3^ spectrum obtained with the ion *m/z* 695.19 fragmented with CID at 35 eV (827.23 CID35 → 695.19 CID35 →). (**D**) MS^4^ spectrum obtained with fragmentation of the ion *m/z* 551.15 with CID at 35 eV (827.23 CID35 → 695.19 CID35 → 551.15 CID35→). Core ions are highlighted in black.

**Table 1 molecules-29-05485-t001:** Chromatographic and mass spectrometry data for betalains from *Hylocereus costaricensis*.

**#**	Compound ^A^	Retention Time (Rt)	Relative Rt from Betanin	Chemical Formula	Theoretical *m/z* [M+H]^+^	Observed *m/z* [M+H]^+^	Mass Accuracy (ppm)	Fragments	Reference ^B^
**1**	Betalamic acid	3.23	0.94	C_9_H_9_NO_5_	212.0553	212.0545	−3.77	194.04, 166.05, 148.04, 138.05, 120.04, 106.03	[15]
**Betanin-type**									
**2**	Prebetanin	3.19	0.93	C_24_H_26_N_2_O_16_S	631.1076	631.1052	−3.8	389.09, 345.10, 343.09, 299.10, 297.08, 281.09, 269.09, 255.11, 253.09, 194.04, 166.05, 178.05, 176.07, 166.05, 150.05, 138.05, 132.04, 106.6	[57,58]
**3**	Betanin	3.43	1	C_24_H_26_N_2_O_13_	551.1508	551.1483	−4.54	389.09, 345.10, 343.09, 299.10, 297.08, 281.09, 269.09, 255.11, 253.09, 194.04, 166.05, 178.05, 176.07, 166.05, 150.05, 138.05, 132.04, 106.6	[9,59]
**4**	Betanidin	3.44	1	C_18_H_16_N_2_O_8_	389.0979	389.0964	−3.86	389.09, 345.10, 343.09, 299.10, 297.08, 281.09, 269.09, 255.11, 253.09, 194.04, 166.05, 178.05, 176.07, 166.05, 150.05, 138.05, 132.04, 106.6	[60]
**5**	2-decarboxy-xanbetanin *	3.62	1.06	C_23_H_24_N_2_O_11_	505.1453	505.1429	−4.75	343.09, 299.10, 297.08, 281.09, 269.09, 255.08, 253.09, 194.04, 178.05, 176.07, 150.05, 138.05, 132.04, 106.06	[61]
**6**	17-decarboxy-betanin	5.83	1.7	C_23_H_26_N_2_O_11_	507.1609	507.1591	−3.55	345.10, 299.10, 297.08, 281.09, 255.11, 253.09, 194.04, 178.05, 176.07, 150.05, 138.05, 132.04, 106.06	[23,49,62]
**7**	Isoprebetanin	6.01	1.75	C_24_H_26_N_2_O_16_S	631.1076	631.1047	−4.6	389.09, 345.10, 343.09, 299.10, 297.08, 281.09, 269.09, 255.11, 253.09, 194.04, 166.05, 178.05, 176.07, 166.05, 150.05, 138.05, 132.04, 106.6	
**8**	Isobetanin	6.59	1.92	C_24_H_26_N_2_O_13_	551.1508	551.1481	−4.9	389.09, 345.10, 343.09, 299.10, 297.08, 281.09, 269.09, 255.11, 253.09, 194.04, 166.05, 178.05, 176.07, 166.05, 150.05, 138.05, 132.04, 106.6	[9]
**9**	2-decarboxy-isoxanbetanin *	6.6	1.92	C_23_H_24_N_2_O_11_	505.1453	505.1433	−3.96	343.09, 299.10, 297.08, 281.09, 269.09, 255.08, 253.09, 194.04, 178.05, 176.07, 150.05, 138.05, 132.04, 106.06	[22]
**10**	2,17-bidecarboxy-xanbetanin *	7.61	2.22	C_22_H_24_N_2_O_9_	461.1555	461.1534	−4.55	343.09, 299.10, 297.08, 281.09, 269.09, 255.08, 253.09, 194.04, 178.05, 176.07, 150.05, 138.05, 132.04, 106.06	[22,51]
**11**	17-decarboxy-isobetanin	7.69	2.24	C_23_H_26_N_2_O_11_	507.1609	507.1587	−4.34	345.10, 299.10, 297.08, 281.09, 255.11, 253.09, 194.04, 178.05, 176.07, 150.05, 138.05, 132.04, 106.06	
**12**	Phyllocactin	7.71	2.25	C_27_H_28_N_2_O_16_	637.1512	637.1497	−2.35	389.09, 345.10, 343.09, 299.10, 297.08, 281.09, 269.09, 255.11, 253.09, 194.04, 166.05, 178.05, 176.07, 166.05, 150.05, 138.05, 132.04, 106.6	[9,63]
**13**	17-decarboxy-phyllocactin	8.01	2.34	C_26_H_28_N_2_O_14_	593.1613	593.1591	−3.71	345.10, 299.10, 297.08, 281.09, 255.11, 253.09, 194.04, 178.05, 176.07, 150.05, 138.05, 132.04, 106.06	[22,23]
**14**	15-decarboxy-betanin	8.29	2.42	C_23_H_26_N_2_O_11_	507.1609	507.1591	−3.55	345.10, 299.10, 297.08, 281.09, 255.11, 253.09, 194.04, 178.05, 176.07, 150.05, 138.05, 132.04, 106.06	[49,64]
**15**	Neobetanin	8.31	2.42	C_24_H_24_N_2_O_13_	549.1351	549.1336	−2.73	387.07, 341.07, 313.08, 299.10, 299.06, 287.08, 281.09, 269.09, 255.08, 253.09, 194.04, 178.05, 176.07, 166.05, 150.05, 132.04, 106.06	[51]
**16**	17-decarboxy-neobetanin *	8.39	2.45	C_23_H_24_N_2_O_11_	505.1453	505.1438	−2.97	343.09, 299.10, 297.08, 281.09, 269.09, 255.08, 253.09, 194.04, 178.05, 176.07, 150.05, 138.05, 132.04, 106.06	[22]
**17**	2-decarboxy-neobetanin	8.4	2.45	C_23_H_24_N_2_O_11_	505.1453	505.1432	−4.16	343.09, 299.10, 297.08, 281.09, 269.09, 255.08, 253.09, 194.04, 178.05, 176.07, 150.05, 138.05, 132.04, 106.06	[47]
**18**	Hylocerenin	8.39	2.45	C_30_H_34_N_2_O_17_	695.193	695.1899	−4.46	389.09, 345.10, 343.09, 299.10, 297.08, 281.09, 269.09, 255.11, 253.09, 194.04, 166.05, 178.05, 176.07, 166.05, 150.05, 138.05, 132.04, 106.6	[9,65]
**19**	Lampranthin-I *	8.41	2.45	C_33_H_32_N_2_O_15_	697.1875	697.1861	−2.01	389.09, 345.10, 343.09, 299.10, 297.08, 281.09, 269.09, 255.11, 253.09, 194.04, 166.05, 178.05, 176.07, 166.05, 150.05, 138.05, 132.04, 106.6	[66]
**20**	Isophyllocactin	8.46	2.47	C_27_H_28_N_2_O_16_	637.1512	637.1491	−3.3	389.09, 345.10, 343.09, 299.10, 297.08, 281.09, 269.09, 255.11, 253.09, 194.04, 166.05, 178.05, 176.07, 166.05, 150.05, 138.05, 132.04, 106.6	
**21**	17-decarboxy-hylocerenin *	8.72	2.54	C_29_H_34_N_2_O_15_	651.2032	651.2012	−3.07	345.10, 299.10, 297.08, 281.09, 255.11, 253.09, 194.04, 178.05, 176.07, 150.05, 138.05, 132.04, 106.06	[22,23]
**22**	2-decarboxy-betanin	8.83	2.57	C_23_H_26_N_2_O_11_	507.1609	507.1596	−2.56	345.10, 299.10, 297.08, 281.09, 255.11, 253.09, 194.04, 178.05, 176.07, 150.05, 138.05, 132.04, 106.06	[22,23]
**23**	Isolampranthin-I	8.85	2.58	C_33_H_32_N_2_O_15_	697.1875	697.1854	−3.01	389.09, 345.10, 343.09, 299.10, 297.08, 281.09, 269.09, 255.11, 253.09, 194.04, 166.05, 178.05, 176.07, 166.05, 150.05, 138.05, 132.04, 106.6	
**24**	Isohylocerenin	8.88	2.59	C_30_H_34_N_2_O_17_	695.193	695.1909	−3.02	389.09, 345.10, 343.09, 299.10, 297.08, 281.09, 269.09, 255.11, 253.09, 194.04, 166.05, 178.05, 176.07, 166.05, 150.05, 138.05, 132.04, 106.6	[22]
**25**	17-decarboxy-isophyllocactin *	9.21	2.69	C_26_H_28_N_2_O_14_	593.1613	593.1588	−4.21	345.10, 299.10, 297.08, 281.09, 255.11, 253.09, 194.04, 178.05, 176.07, 150.05, 138.05, 132.04, 106.06	
**26**	17-decarboxy-isohylocerenin *	9.26	2.7	C_29_H_34_N_2_O_15_	651.2032	651.2009	−3.53	345.10, 299.10, 297.08, 281.09, 255.11, 253.09, 194.04, 178.05, 176.07, 150.05, 138.05, 132.04, 106.06	
**27**	2-descarboxy-phyllocactin	9.38	2.73	C_26_H_28_N_2_O_14_	593.1613	593.1587	−4.38	345.10, 299.10, 297.08, 281.09, 255.11, 253.09, 194.04, 178.05, 176.07, 150.05, 138.05, 132.04, 106.06	[22,23]
**28**	2-descarboxy-isophyllocactin *	9.41	2.74	C_26_H_28_N_2_O_14_	593.1613	593.1589	−4.05	345.10, 299.10, 297.08, 281.09, 255.11, 253.09, 194.04, 178.05, 176.07, 150.05, 138.05, 132.04, 106.06	
**29**	2-decarboxy-xanneobetanin *	10.06	2.93	C_23_H_22_N_2_O_11_	503.1296	503.1277	−3.78	341.07, 327.06, 313.08, 295.07, 277.07, 267.07, 253.06, 251.08, 221.07, 195.09, 132.04, 106.06	[51]
**30**	Neophyllocactin *	10.13	2.95	C_27_H_26_N_2_O_16_	635.1355	635.1327	−4.41	387.07, 341.07, 313.08, 299.10, 299.06, 287.08, 281.09, 269.09, 255.08, 253.09, 194.04, 178.05, 176.07, 166.05, 150.05, 132.04, 106.06	[22]
**31**	2,15,17-tridecarboxy-neobetanin *	10.43	3.04	C_21_H_24_N_2_O_7_	417.1656	417.1648	−1.92	417.17, 349.11, 271.09, 255.11, 159.04, 130.03	[47]
**32**	17-decarboxy-neophyllocactin *	10.52	3.07	C_26_H_26_N_2_O_14_	591.1457	591.1432	−4.23	343.09, 299.10, 297.08, 281.09, 269.09, 255.08, 253.09, 194.04, 178.05, 176.07, 150.05, 138.05, 132.04, 106.06	[22]
**33**	Neohylocerenin	10.89	3.17	C_30_H_32_N_2_O_17_	693.1774	693.1758	−2.31	387.07, 341.07, 313.08, 299.10, 299.06, 287.08, 281.09, 269.09, 255.08, 253.09, 194.04, 178.05, 176.07, 166.05, 150.05, 132.04, 106.06	
**34**	17-decarboxy-neohylocerenin *	11.04	3.22	C_29_H_32_N_2_O_15_	649.1875	649.1852	−3.54	343.09, 299.10, 297.08, 281.09, 269.09, 255.08, 253.09, 194.04, 178.05, 176.07, 150.05, 138.05, 132.04, 106.06	[22]
**35**	2,17-bidecarboxy-xanneobetanin *	11.14	3.25	C_22_H_22_N_2_O_9_	459.1398	459.1383	−3.27	297.08, 269.09, 251.08, 223.08, 195.09, 133.08	[51]
**36**	Lampranthin II	11.01	3.21	C_34_H_34_N_2_O_16_	727.1981	727.1956	−3.44	389.09, 345.10, 343.09, 299.10, 297.08, 281.09, 269.09, 255.11, 253.09, 194.04, 166.05, 178.05, 176.07, 166.05, 150.05, 138.05, 132.04, 106.6	[59,66,67]
**37**	Isolampranthin II	11.34	3.31	C_34_H_34_N_2_O_16_	727.1981	727.1948	−4.54	389.09, 345.10, 343.09, 299.10, 297.08, 281.09, 269.09, 255.11, 253.09, 194.04, 166.05, 178.05, 176.07, 166.05, 150.05, 138.05, 132.04, 106.6	
**38**	2-decarboxy-neohylocerenin	11.45	3.34	C_29_H_32_N_2_O_15_	649.1875	649.1848	−4.16	343.09, 299.10, 297.08, 281.09, 269.09, 255.08, 253.09, 194.04, 178.05, 176.07, 150.05, 138.05, 132.04, 106.06	[22]
**39**	2-decarboxy-neophyllocactin *	11.62	3.39	C_26_H_26_N_2_O_14_	591.1457	591.1438	−3.21	343.09, 299.10, 297.08, 281.09, 269.09, 255.08, 253.09, 194.04, 178.05, 176.07, 150.05, 138.05, 132.04, 106.06	[22]
**40**	Neolampranthin II	11.64	3.39	C_34_H_32_N_2_O_16_	725.1825	725.1794	−4.27	387.07, 341.07, 313.08, 299.10, 299.06, 287.08, 281.09, 269.09, 255.08, 253.09, 194.04, 178.05, 176.07, 166.05, 150.05, 132.04, 106.06	
**41**	2,15,17-tridecarboxy-xanneobetanin *	11.94	3.48	C_21_H_22_N_2_O_7_	415.1499	415.1497	−0.48	415.14, 355.12, 347.09, 285.09, 185.04, 143.02	[51]
**Melocactin-type**									
**42**	Melocactin	4.81	1.4	C_30_H_36_N_2_O_18_	713.2036	713.2005	−4.35	389.09, 345.10, 343.09, 299.10, 297.08, 281.09, 269.09, 255.11, 253.09, 194.04, 166.05, 178.05, 176.07, 166.05, 150.05, 138.05, 132.04, 106.6	[68,69]
**43**	Isomelocactin	7.18	2.09	C_30_H_36_N_2_O_18_	713.2036	713.2009	−3.79	389.09, 345.10, 343.09, 299.10, 297.08, 281.09, 269.09, 255.11, 253.09, 194.04, 166.05, 178.05, 176.07, 166.05, 150.05, 138.05, 132.04, 106.6	[68]
**Apiocactin-type**									
**44**	Apiocactin	7.21	2.1	C_29_H_34_N_2_O_17_	683.193	683.1904	−3.81	389.09, 345.10, 343.09, 299.10, 297.08, 281.09, 269.09, 255.11, 253.09, 194.04, 166.05, 178.05, 176.07, 166.05, 150.05, 138.05, 132.04, 106.6	[69]
**45**	Isoapiocactin	7.91	2.31	C_29_H_34_N_2_O_17_	683.193	683.1911	−2.78	389.09, 345.10, 343.09, 299.10, 297.08, 281.09, 269.09, 255.11, 253.09, 194.04, 166.05, 178.05, 176.07, 166.05, 150.05, 138.05, 132.04, 106.6	
**46**	2′-O-apiosyl-phyllocactin *	8.31	2.42	C_32_H_36_N_2_O_20_	769.1934	769.1912	−2.86	389.09, 345.10, 343.09, 299.10, 297.08, 281.09, 269.09, 255.11, 253.09, 194.04, 166.05, 178.05, 176.07, 166.05, 150.05, 138.05, 132.04, 106.6	[65,69]
**47**	2′-O-apiosyl-isophyllocactin *	8.76	2.55	C_32_H_36_N_2_O_20_	769.1934	769.1903	−4.03	389.09, 345.10, 343.09, 299.10, 297.08, 281.09, 269.09, 255.11, 253.09, 194.04, 166.05, 178.05, 176.07, 166.05, 150.05, 138.05, 132.04, 106.6	[65]
**48**	17-decarboxy-2′-O-β-apiosyl-phyllocactin *	9.89	2.88	C_31_H_36_N_2_O_18_	725.2036	725.2011	−3.45	345.10, 299.10, 297.08, 281.09, 255.11, 253.09, 194.04, 178.05, 176.07, 150.05, 138.05, 132.04, 106.06	
**49**	17-decarboxy-2′-O-β-apiosyl-isophyllocactin *	10.19	2.97	C_31_H_36_N_2_O_18_	725.2036	725.2007	−4.00	345.10, 299.10, 297.08, 281.09, 255.11, 253.09, 194.04, 178.05, 176.07, 150.05, 138.05, 132.04, 106.06	[65]
**Gomphrenin-type**									
**50**	Gomphrenin-I	4.06	1.18	C_24_H_26_N_2_O_13_	551.1508	551.1481	−4.9	389.09, 345.10, 343.09, 299.10, 297.08, 281.09, 269.09, 255.11, 253.09, 194.04, 166.05, 178.05, 176.07, 166.05, 150.05, 138.05, 132.04, 106.6	[59,70]
**51**	2-decarboxy-xangomphrenin *	6.71	1.96	C_23_H_24_N_2_O_11_	505.1453	505.1431	−4.36	343.09, 299.10, 297.08, 281.09, 269.09, 255.08, 253.09, 194.04, 178.05, 176.07, 150.05, 138.05, 132.04, 106.06	[71]
**52**	Isogomphrenin-I	6.87	2	C_24_H_26_N_2_O_13_	551.1508	551.1486	−3.99	389.09, 345.10, 343.09, 299.10, 297.08, 281.09, 269.09, 255.11, 253.09, 194.04, 166.05, 178.05, 176.07, 166.05, 150.05, 138.05, 132.04, 106.6	[72]
**53**	Neogomphrenin	8.58	2.5	C_24_H_24_N_2_O_13_	549.1351	549.1327	−4.37	387.07, 341.07, 313.08, 299.10, 299.06, 287.08, 281.09, 269.09, 255.08, 253.09, 194.04, 178.05, 176.07, 166.05, 150.05, 132.04, 106.06	
**54**	Gomphrenin-IV	10.91	3.18	C_35_H_36_N_2_O_17_	757.2087	757.2065	−2.91	389.09, 345.10, 343.09, 299.10, 297.08, 281.09, 269.09, 255.11, 253.09, 194.04, 166.05, 178.05, 176.07, 166.05, 150.05, 138.05, 132.04, 106.6	[50,73]
**55**	Isogomphrenin-IV	11.16	3.25	C_35_H_36_N_2_O_17_	757.2087	757.2071	−2.11	389.09, 345.10, 343.09, 299.10, 297.08, 281.09, 269.09, 255.11, 253.09, 194.04, 166.05, 178.05, 176.07, 166.05, 150.05, 138.05, 132.04, 106.6	
**56**	Gomphrenin-III	11.33	3.3	C_34_H_34_N_2_O_16_	727.1981	727.1949	−4.4	389.09, 345.10, 343.09, 299.10, 297.08, 281.09, 269.09, 255.11, 253.09, 194.04, 166.05, 178.05, 176.07, 166.05, 150.05, 138.05, 132.04, 106.6	[50,72]
**57**	Isogomphrenin-III	11.62	3.39	C_34_H_34_N_2_O_16_	727.1981	727.1957	−3.3	389.09, 345.10, 343.09, 299.10, 297.08, 281.09, 269.09, 255.11, 253.09, 194.04, 166.05, 178.05, 176.07, 166.05, 150.05, 138.05, 132.04, 106.6	
**58**	Neogomphrenin IV	11.59	3.38	C_35_H_34_N_2_O_17_	755.193	755.1914	−2.12	387.07, 341.07, 313.08, 299.10, 299.06, 287.08, 281.09, 269.09, 255.08, 253.09, 194.04, 178.05, 176.07, 166.05, 150.05, 132.04, 106.06	
**59**	Neogomphrenin-III	11.92	3.48	C_34_H_32_N_2_O_16_	725.1825	725.1798	−3.72	387.07, 341.07, 313.08, 299.10, 299.06, 287.08, 281.09, 269.09, 255.08, 253.09, 194.04, 178.05, 176.07, 166.05, 150.05, 132.04, 106.06	
**Glabranin-type**									
**60**	Glabranin	3.15	0.92	C_30_H_36_N_2_O_18_	713.2036	713.2015	−2.94	389.09, 345.10, 343.09, 299.10, 297.08, 281.09, 269.09, 255.11, 253.09, 194.04, 166.05, 178.05, 176.07, 166.05, 150.05, 138.05, 132.04, 106.6	[74]
**61**	Coumglabranin *	10.59	3.09	C_39_H_42_N_2_O_20_	859.2403	859.2377	−3.03	389.09, 345.10, 343.09, 299.10, 297.08, 281.09, 269.09, 255.11, 253.09, 194.04, 166.05, 178.05, 176.07, 166.05, 150.05, 138.05, 132.04, 106.6	[74]
**62**	Isocoumglabranin *	10.85	3.16	C_39_H_42_N_2_O_20_	859.2403	859.2371	−3.72	389.09, 345.10, 343.09, 299.10, 297.08, 281.09, 269.09, 255.11, 253.09, 194.04, 166.05, 178.05, 176.07, 166.05, 150.05, 138.05, 132.04, 106.6	
**63**	Neocoumglabranin *	11.29	3.29	C_39_H_40_N_2_O_20_	857.2247	857.2208	−4.55	387.07, 341.07, 313.08, 299.10, 299.06, 287.08, 281.09, 269.09, 255.08, 253.09, 194.04, 178.05, 176.07, 166.05, 150.05, 132.04, 106.06	
**Unknown type**									
**64**	Unknown-1 *	8.83	2.57	C_35_H_42_N_2_O_21_	827.2353	827.2316	−4.47	389.09, 345.10, 343.09, 299.10, 297.08, 281.09, 269.09, 255.11, 253.09, 194.04, 166.05, 178.05, 176.07, 166.05, 150.05, 138.05, 132.04, 106.6	
**65**	Unknown-2 *	9.23	2.69	C_35_H_42_N_2_O_21_	827.2353	827.2328	−3.02	389.09, 345.10, 343.09, 299.10, 297.08, 281.09, 269.09, 255.11, 253.09, 194.04, 166.05, 178.05, 176.07, 166.05, 150.05, 138.05, 132.04, 106.6	
**66**	Unknown-3 *	10.27	2.99	C_34_H_42_N_2_O_19_	783.2455	783.2422	−4.21	345.10, 299.10, 297.08, 281.09, 255.11, 253.09, 194.04, 178.05, 176.07, 150.05, 138.05, 132.04, 106.06	
**67**	Unknown-4 *	10.54	3.07	C_34_H_42_N_2_O_19_	783.2455	783.2428	−3.45	345.10, 299.10, 297.08, 281.09, 255.11, 253.09, 194.04, 178.05, 176.07, 150.05, 138.05, 132.04, 106.06	
**Betaxanthin**									
**Positively charged-type s**									
**68**	Histamine-bx *	9.04	2.64	C_14_H_16_N_4_O_4_	305.1244	305.1232	−3.93	287.12, 261.13, 256.89, 211.07, 194.04, 166.05, 150.05, 138.05, 132.04, 130.05, 106.06	
**Polar uncharged-type**									
**69**	Glutamine-bx *	1.2	0.35	C_14_H_17_N_3_O_7_	340.1139	340.1129	−2.94	323.08, 277.08, 249.08, 233.09, 231.07, 211.07, 194.04, 166.05, 150.05, 138.05, 132.04, 130.05, 106.06	[15]
**70**	Threonine-bx *	1.52	0.44	C_13_H_16_N_2_O_7_	313.103	313.1022	−2.56	269.09, 267.09, 211.07, 194.04, 166.05, 150.05, 138.05, 132.04, 130.05, 106.06	[15]
**71**	Serine-bx *	4.77	1.39	C_12_H_14_N_2_O_7_	299.0874	299.0861	−4.35	255.09, 253.09, 211.07, 194.04, 166.05, 150.05, 138.05, 132.04, 130.05, 106.06	[15]
**Hydrophobic-type**									
**72**	Glycine-bx *	6.49	1.89	C_11_H_12_N_2_O_6_	269.0768	269.0758	−3.72	331.12, 239.11, 285.12, 283.10, 239.11, 211.07, 194.04, 166.05, 150.05, 138.05, 132.04, 130.05, 106.06	[15]
**73**	Methionine-bx *	7.3	2.13	C14H18N2O6S	343.0958	343.0943	−4.37	315.09, 269.09, 211.07, 194.04, 166.05, 150.05, 138.05, 132.04, 130.05, 106.06	[15]
**74**	Valine-bx *	7.97	2.32	C14H18N2O6	311.1238	311.1229	−2.89	267.13, 265.11, 237.12, 221.12, 219.11, 211.07, 194.04, 166.05, 150.05, 138.05, 132.04, 130.05, 106.06	[15]
**75**	Alanine-bx *	8.16	2.38	C12H14N2O6	283.0925	283.0915	−3.53	237.08, 211.07, 194.04, 166.05, 150.05, 138.05, 132.04, 130.05, 106.06	[15]
**75**	Proline-bx *	9.02	2.63	C_14_H_16_N_2_O_6_	309.1081	309.1068	−4.21	265.11, 263.10, 235.1, 219.11, 217.09, 211.07, 194.04, 166.05, 150.05, 138.05, 132.04, 130.05, 106.06	[15]
**77**	Isoleucine-bx *	9.87	2.88	C_15_H_20_N_2_O_6_	325.1394	325.1384	−3.08	279.13, 251.13, 235.14, 233.12, 211.07, 194.04, 166.05, 150.05, 138.05, 132.04, 130.05, 106.06	[15]
**78**	Leucine-bx *	10.01	2.92	C_15_H_20_N_2_O_6_	325.1394	325.1385	−2.77	281.14, 279.13, 251.13, 235.14, 233.12, 211.07, 194.04, 166.05, 150.05, 138.05, 132.04, 130.05, 106.06	[15]
**79**	Tryptophan-bx *	10.12	2.95	C_20_H_19_N_3_O_6_	398.1347	398.1331	−4.02	269.07, 223.07, 211.07, 194.04, 166.05, 150.05, 138.05, 132.04, 130.05, 106.06	[15]
**80**	Phenylalanine-bx *	10.15	2.96	C_18_H_18_N_2_O_6_	359.1238	359.1225	−3.62	315.13, 313.11, 269.12, 267.11, 223.12, 211.07, 194.04, 166.05, 150.05, 138.05, 132.04, 130.05, 106.06	[15]
**81**	g-aminobutyric acid-bx *	2.82	0.82	C_13_H_16_N_2_O_6_	297.1081	297.1069	−4.04	253.11, 251.10, 233.09, 211.07, 194.04, 166.05, 150.05, 138.05, 132.04, 130.05, 106.06	
**82**	Dopamine-bx *	7.67	2.24	C_17_H_18_N_2_O_6_	347.1238	347.1226	−3.46	255.11, 211.07, 194.04, 166.05, 150.05, 138.05, 132.04, 130.05, 106.06	[15]
**83**	Tyramine-bx *	8.64	2.52	C_17_H_18_N_2_O_5_	331.1288	331.1274	−4.23	287.13, 285.12, 239.11, 211.07, 194.04, 166.05, 150.05, 138.05, 132.04, 130.05, 106.06	
**84**	3-methoxy-tyramine-bx *	8.98	2.62	C_18_H_20_N_2_O_6_	361.1394	361.1383	−3.05	315.13, 269.12, 211.07, 194.04, 166.05, 150.05, 138.05, 132.04, 130.05, 106.06	
**85**	5-hydroxynorvaline-bx *	9.89	2.88	C_14_H_18_N_2_O_7_	327.1187	327.1179	−2.45	283.15, 237.15, 211.07, 194.04, 166.05, 150.05, 138.05, 132.04, 130.05, 106.06	
**86**	Methionine sulfoxide-bx *	10.09	2.94	C_14_H_18_N_2_O_7_S	359.0907	359.0893	−3.9	267.11, 223.12, 211.07, 194.04, 166.05, 150.05, 138.05, 132.04, 130.05, 106.06	

^A^ The prefix “xan” is equivalent to 2,3-dehydro, while “neo” is equivalent to 14, 15-dehydro, as described by Starzak et al. [75]. * Tentatively identified. ^B^ A reference to the identified betalain.

## Data Availability

Data are contained within the article and Appendix A.

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
