# Peer review of "In Silico Mass Spectrometric Fragmentation and Liquid Chromatography with Tandem Mass Spectrometry (LC-MS/MS) Betalainic Fingerprinting: Identification of Betalains in Red Pitaya"

_molecules, 2024, doi:10.3390/molecules29225485_

Round 1
Reviewer 1 Report
Comments and Suggestions for Authors
In general, the idea of the manuscript is interesting, however, many complex betacyanins of Bougainvillea glabra are omitted in the Introduction and Results sections, so this report describes basic structures. Indeed, more complex structures still await studies. The application of spectrophotometry and LCMS just to this case of chosen red beet and pitaya seems justified but not all data including those from Vis detection are presented to support or “double check” the MS results. The conclusions are sound but could be supported by additional chromatograms. This would be a good starting point for analyses of other pitaya varietes - it is unknown whether other varieties will show the spectrophotometric and LC-MS profiles similar to these described in this report.
Detailed comments:
Lines 91-108 - Currently, by searching through the last decade literature, you can see that there are many more than 34 betacyanins detected in plants by at least LC-DAD-MS techniques – please, refer to ref. [42] - which unequivocally indicated the presence of these pigments.
Taking into account Bougainvillea glabra enormous number of ca. 146 betacyanins [“Ion-pair high-speed countercurrent chromatography in fractionation of a high-molecular weight variation of acyl-oligosaccharide linked betacyanins from purple bracts of Bougainvillea glabra, Journal of Chromatography B, 878 (2010) 538–550”] it should be stated that together over 180 betacyanins [ref. 42] were detected even if many of them still await detailed structural studies.
In addition their m/z is frequently higher that 1000.
Lines 20, 27, 39, 57, 62, 90, 86, 112, 133 , 189, 208, 266, 286, 398, 399 and 415 should be “betalain” or “betalainic” or “betalains’ “ instead of betalains because here this word is used in possesive form
Lines 134, 157. – similarly to above, please change “betacyanins” and “betaxanthins” into appropriate forms
Line 150. Please, correct the m/z of melocactin-type
Line 175. It should be “betanidin/isobetanidin”
Line 185. “m/z 345.10 due to loss of a carboxyl group….” But not carbonyl
Line 186. “m/z 343.09 due to loss of CO2 and H2, either HCOOH, either CO and H2O – this is still not well recognized in the literature,”
Line 186. Should be “other” not another
Line 197. Why betalains? These ions are specifically for betacyanins.
Line 233. 2-decarboxy-neobetanin
Lines 266-281. A figure with chromatograms indicating ion traces of isomeric structures as well as Vis detection at selected wavelengths for comparison would clarify the whole profile of the pigments from pitaya extract indicating the major and minor components.
Line 385. What do you mean by “for all known betalains”? Most of Bougainvillea glabra enormous number of ca. 146 betacyanins is not mentioned in this report
Line 401. should be “rapid”
Line 403-404. should be “...increase of annotation quality of metabolite identification…”.
Table 1.:
- To make it simpler, you can write “xan” instead of “2,3-dehydro-“ in all the names. The names with text “2,3-dehydro-“ can be deleted instead. You can also explain the meaning of “xan” referring to appropriate references under the Table. In the same way, you can explain “neo”.
- Some of betalains and their derivatives are fully identified by NMR and LCMS, and some only tentatively identified by LCMS, therefore, this fact should be mentioned wherever necessary (Table 1, text), please, put asterisk behind appropriate compound which was not identified by NMR in the first column and write “Tentatively identified” below the Table. This way, you can confront the results obtained by PDA detection with with results from the MassFrontier software.
- In addition, it would be valuable to insert absorption maxima of the compounds into a new column in the Table because they are very useful in recognizing a group of betalainic derivatives – they are very characteristic for each group.
Fig 3A. The bond stereo at C-15 should have opposite direction
Fig 4A and B. – In the case of ion m/z 343 chemical structure, please, explain why the opening of the 5-atom ring is proposed? Similarly to Line 186, we don’t know which mechanism ion m/z 343 can be considered.
Fig. 6 – Can you propose the fragment structures in the Figure 6 and draw them similarly to Fig. 4.?
Author Response
Reviewer #1
Dear Reviewer, we are grateful for your valuable feedback on our manuscript. We appreciate them, and they have been great in improving the paper. We carefully reviewed the comments made and answered as best we could.
Reviewer 1
In general, the idea of the manuscript is interesting, however, many complex betacyanins of Bougainvillea glabra are omitted in the Introduction and Results sections, so this report describes basic structures. Indeed, more complex structures still await studies. The application of spectrophotometry and LCMS just to this case of chosen red beet and pitaya seems justified but not all data including those from Vis detection are presented to support or “double check” the MS results. The conclusions are sound but could be supported by additional chromatograms. This would be a good starting point for analyses of other pitaya varietes - it is unknown whether other varieties will show the spectrophotometric and LC-MS profiles similar to these described in this report.
Detailed comments:
Comment 1: Lines 91-108 - Currently, by searching through the last decade literature, you can see that there are many more than 34 betacyanins detected in plants by at least LC-DAD-MS techniques – please, refer to ref. [42] - which unequivocally indicated the presence of these pigments.
Taking into account Bougainvillea glabra enormous number of ca. 146 betacyanins [“Ion-pair high-speed countercurrent chromatography in fractionation of a high-molecular weight variation of acyl-oligosaccharide linked betacyanins from purple bracts of Bougainvillea glabra, Journal of Chromatography B, 878 (2010) 538–550”] it should be stated that together over 180 betacyanins [ref. 42] were detected even if many of them still await detailed structural studies.
In addition their m/z is frequently higher that 1000.
Response 1: Thank you so much for your comments. We have added “Journal of Chromatography B, 878 (2010) 538–550” as a new reference in the main text. This new reference highlights the point raised in the revision about the combined number of betacyanins from different species and their molecular weight exceeding 1000 Da. The text that was added appears as:
“and 146 betacyanins, most of which have a high molecular weight of over 1000 Da, in Bougainvillea glabra [42]. Given that some betacyanins are specific to certain species, the combined number of betacyanins from different species exceeds 200 [42, 43], which highlights the wide variety of these pigments.”
Comment 2: Lines 20, 27, 39, 57, 62, 90, 86, 112, 133 , 189, 208, 266, 286, 398, 399 and 415 should be “betalain” or “ac” or “betalains’ “ instead of betalains because here this word is used in possesive form.
Response 2: Thank you so much for your comments. The main text has been corrected.
Comment 3: Lines 134, 157. – similarly to above, please change “betacyanins” and “betaxanthins” into appropriate forms.
Response 3: Thank you so much for your comments. The main text has been corrected.
Comment 4: Line 150. Please, correct the m/z of melocactin-type
Response 4: Thank you so much for your comment. The m/z value has been corrected.
Comment 5: Line 175. It should be “betanidin/isobetanidin”
Response 5: Thank you so much for your comment. The suggestion was introduced.
Comment 6: Line 185. “m/z 345.10 due to loss of a carboxyl group….” But not carbonyl
Response 6: Thank you so much for your comment. The correction was made.
Comment 7: Line 186. “m/z 343.09 due to loss of CO2 and H2, either HCOOH, either CO and H2O – this is still not well recognized in the literature,”
Response 7: Thank you so much for your comment. The correction was made.
Comment 8: Line 186. Should be “other” not another
Response 8: Thank you so much for your comment. The correction was made.
Comment 9: Line 197. Why betalains? These ions are specifically for betacyanins.
Response 9: Thank you so much for your comment. The main reason to suggest that those ions are essential for the identification of betalains instead of only betacyanins came from the last eight ions mentioned. The ions were derived from the core structure of the betalains that are in the pyridine ring. Indeed, it was found that those ions in betaxanthins are recurrent (see Table 1 (Betaxanthins) and last paragraph of section 2.3), for example, the ion with m/z 211.07.
Comment 10: Line 233. 2-decarboxy-neobetanin
Response 10: Thank you so much for your comments. The correction was made.
Comment 11: Lines 266-281. A figure with chromatograms indicating ion traces of isomeric structures as well as Vis detection at selected wavelengths for comparison would clarify the whole profile of the pigments from pitaya extract indicating the major and minor components.
Response 11: Thank you so much for your comments. The point out has been addressed by adding the chromatogram of red pitaya at 540 nm to Figure 7.
Comment 12: Line 385. What do you mean by “for all known betalains”? Most of Bougainvillea glabra enormous number of ca. 146 betacyanins is not mentioned in this report
Response 12: Thank you so much for your comment. Information about betacyanins of high molecular weight from Bougainvillea glabra has now been mentioned in lines 108-112.
Comment 13: Line 401. should be “rapid”
Response 13: Thank you so much for your comment. The correction was made.
Comment 14: Line 403-404. should be “...increase of annotation quality of metabolite identification…”.
Response 14: Thank you so much for your comment. The correction was made.
Comments Table 1.:
Comment 15: To make it simpler, you can write “xan” instead of “2,3-dehydro-“ in all the names. The names with text “2,3-dehydro-“ can be deleted instead. You can also explain the meaning of “xan” referring to appropriate references under the Table. In the same way, you can explain “neo”.
Response 15: However, we include the terms 'xan' and 'neo' under the table. In addition, we have included a reference that provides a detailed description of those terms.
Comment 16: Some of betalains and their derivatives are fully identified by NMR and LCMS, and some only tentatively identified by LCMS, therefore, this fact should be mentioned wherever necessary (Table 1, text), please, put asterisk behind appropriate compound which was not identified by NMR in the first column and write “Tentatively identified” below the Table. This way, you can confront the results obtained by PDA detection with with results from the MassFrontier software.
Response 16: Thank you so much for your comment. We have added the “Tentative identified” below accompanying the Table. Our UHPLC-UV-Vis system is not equipped with DAD or PDA detectors. An additional column was also added to the table with references where the identification was reported by NMR or MS.
Comment 17: In addition, it would be valuable to insert absorption maxima of the compounds into a new column in the Table because they are very useful in recognizing a group of betalainic derivatives – they are very characteristic for each group.
Response 17: Thank you so much for your comment. Absorption data is valuable, but it is not available to us. Our UHPLC-UV-Vis system is not equipped with DAD or PDA detectors. The measurements we have are limited to 540 nm. Due to the single-channel system, we cannot obtain the maximum absorption wavelengths.
Comment 18: Fig 3A. The bond stereo at C-15 should have opposite direction.
Response 18: Thank you so much for your comment. The Figure 3A was double-checked and we confirmed that it is correct. We would like to note that the dashed line and solid line represent the bond stereo at C-15.
Comment 19: Fig 4A and B. – In the case of ion m/z 343 chemical structure, please, explain why the opening of the 5-atom ring is proposed? Similarly to Line 186, we don’t know which mechanism ion m/z 343 can be considered.
Response 19: Thank you so much for your comments. We would like to note that the structures placed on the spectrum for explaining the ions in this work were based on the use of MassFrontier software. Where multiple pathways supporting the mechanism for generating each ion are available. Below is the mechanism that explains the ion with m/z 343.
Comment 20: Fig. 6 – Can you propose the fragment structures in the Figure 6 and draw them similarly to Fig. 4.?
Response 20: Thank you so much for your comment. Structures explaining the product ions were added to Figure 6.
Sincerely,
Dr. Victor Aguilar-Hernández
Unidad de Biología Integrativa
Centro de Investigación Científica de Yucatán
Email: victor.aguilar@cicy.mx

Reviewer 2 Report
Comments and Suggestions for Authors
The manuscript on “In Silico Mass Spectrometry Fragmentation and LC-MS/MS in Betalains: Fingerprint and Identification of Betalains in Red Pitaya” by Araujo-León et al. describes detailed LC-MS/MS analyses of betalains and provides a material that could be utilized as a database. The manuscript is clearly written and has nice results but seems to be a little technical, for example, the novelty and applications of these results could be highlighted more. I suggest here some corrections and, in addition, I would like the authors to add the database material to some already existing open access platform, so that all readers could use it directly instead of a lot of manual handwork to actually use the database from the supplementary material.
Minor comments:
· Abstract could be clarified. It is a little bit hard to follow the work as first, a database of 136 betalains (86+23) is presented, and later, it is said that the use of red pitaya allowed to expand the betalains catalog to 86 (31+36+19).
· Introduction provides sufficient background and includes all relevant references. But what is actually meant by “betalains catalog” and “expansion of the betalains catalog”? Is there a catalog (database) already somewhere or just more identified betalains in the literature?
· Add the charge to the molecular ion [M+H]+ (+ with superscript, throughout the manuscript).
· Some example structures could be added to Figure 1, for example, to show the C-15 core (line 139).
· Check the order of tables in supplementary materials. Table S3 is the first one mentioned in the manuscript text.
· How realistic are the theoretical fragments shown for each betalain in TableS3? For example, for rivinianin, what m/z 630, 629 and 628 could be?
· What does Figure 2 tell us? What is its additional value?
· Regarding identification of betalains in Table 1: 1) Did the polyphenols present, such as betagarin or betavulgarin, affect the analysis and detection of betalains? 2) Did you detect vulgaxanthins I and II? 3) Check the identification of gomphrenin II. The molecular formula does not match with the literature data (C33H32N2O15) 4) Check the identification and naming of all glabranin-type betalains. 5) How can a polar (uncharged-type) betalain (71) elute later in reversed phase chromatography than hydrophobic-type betalains (72,73)?
· Correct Betaxhanitns to Betaxanthins in Table 1.
· Please, modify/clarify the text regarding collision energies used for CID in Figure 4 caption.
· Please, modify Figures 5, 6 and 8 to be similar to Figure 4: add collision energies and structures of fragments on the top of the peaks in mass spectra.
· Please, add the chromatogram measured to Figure 7.
· Please, improve the layout of Table 1. Could it be horizontal to have more room for the title and compound names?
· Please, add all parameters related to MassFrontier analysis. What does “The fragmentation parameters were established to be steric optimal, allowing resonance reactions for electron sharing, charge stabilization, and radical isomerization.” mean? Can you detect radical isomerization for betalains when CID at 50 eV is used?
· Conclusions: it is not now a database, but values in the supplementary material. Could you make it more accessible? For example, by adding the data to an already existing open access platform or to databases of Thermo Scientific (e.g. to be used via CompoundDiscoverer softaware)? You have made a huge and careful work but it is quite hard for the others to take an advantage of that.
In general, I do not feel that I’m qualified to access the quality of English but I would change some of the words used in the manuscript, like in the title: mass spectrometry fragmentation -> mass spectrometric fragmentation and fingerprint (the end result) -> fingerprinting (the technique).
Author Response
Reviewer #2
Dear Reviewer, we are grateful for your valuable feedback on our manuscript. We appreciate them, and they have been great in improving the paper. We carefully reviewed the comments made and answered as best we could.
Reviewer 2:
The manuscript on “In Silico Mass Spectrometry Fragmentation and LC-MS/MS in Betalains: Fingerprint and Identification of Betalains in Red Pitaya” by Araujo-León et al. describes detailed LC-MS/MS analyses of betalains and provides a material that could be utilized as a database. The manuscript is clearly written and has nice results but seems to be a little technical, for example, the novelty and applications of these results could be highlighted more. I suggest here some corrections and, in addition, I would like the authors to add the database material to some already existing open access platform, so that all readers could use it directly instead of a lot of manual handwork to actually use the database from the supplementary material.
Minor comments:
Comment 1: Abstract could be clarified. It is a little bit hard to follow the work as first, a database of 136 betalains (86+23) is presented, and later, it is said that the use of red pitaya allowed to expand the betalains catalog to 86 (31+36+19).
Response 1: Thank you so much for your comment. The passage about betalains in red pitaya was reorganized to prevent confusion.
Comment 2: Introduction provides sufficient background and includes all relevant references. But what is actually meant by “betalains catalog” and “expansion of the betalains catalog”? Is there a catalog (database) already somewhere or just more identified betalains in the literature?
Response 2: Thank you so much for your comments. There is no database of betalains in existence. It should be useful to generate a database of betalains that is manageable for non-specialized users in mass spectrometry. This action will require assistance from a bioinformatics specialist to generate the facility. We refer to a catalog as a set of betalains either in the literature or in this work, and it can be expanded by adding new elements through the 'expansion of betalains catalog'.
Comment 3: Add the charge to the molecular ion [M+H]+ (+ with superscript, throughout the manuscript).
Response 3: Thank you so much for your comment. The text was revised to correct the molecular ion charge.
Comment 4: Some example structures could be added to Figure 1, for example, to show the C-15 core (line 139).
Response 4: Given that in Figure 3A, the C15 is displayed in betanin and isobetanin this is an example of the C-15 core. Figure 3A is mentioned in line 143.
Comment 5: Check the order of tables in supplementary materials. Table S3 is the first one mentioned in the manuscript text.
Response 5: Thank you so much for your comment. The supplementary material has been reordered according to the main text.
Comment 6: How realistic are the theoretical fragments shown for each betalain in TableS3? For example, for rivinianin, what m/z 630, 629 and 628 could be?
Response 6: Thank you so much for your comment. We would like to point out that the ions in Table S3 (now Table S1) were predicted by MassFrontier, which uses chemical fragmentation rules and generates chemical adducts and plausible mechanisms to generate them. In principle, there are all possible ions, but some may not be visible in spectra due to, for instance, the detector used in the mass spectrometer or instability, among other factors. In the case of Rivinianin, the ions pointed out are related to charge and radical rearrangements. Definitively, more research with individual betalains is required to highlight the ions that appear in the spectra and to understand the complete fragmentation paths in betalains, but we think that supplementary information is a great advance, especially for the scientific community without access to the MassFrontier software. We detected some of those predicted ions using the orbitrap detector and extracts from beetroot and red pitaya.
Comment 7: What does Figure 2 tell us? What is its additional value?
Response 7: Thank you so much for your comment. In Figure 2, we would like to display the diversity of betalains in terms of molecular ion size of each betalain and its relationship with the classes in the main two classes, betaxanthins and betacyanins. This is an innovative and new representation of the diversity of betalains. It also emphasizes that betacyanin classes have a betacyanin core type framework, from which the addition of glycoside groups increases the molecular size, while the loss of carboxylic acid and H2 decreases it.
Comment 8: Regarding identification of betalains in Table 1: 1) Did the polyphenols present, such as betagarin or betavulgarin, affect the analysis and detection of betalains? 2) Did you detect vulgaxanthins I and II? 3) Check the identification of gomphrenin II. The molecular formula does not match with the literature data (C33H32N2O15) 4) Check the identification and naming of all glabranin-type betalains. 5) How can a polar (uncharged-type) betalain (71) elute later in reversed phase chromatography than hydrophobic-type betalains (72,73)?
Response 8: Thank you so much for your comments.
Comment 1) Did the polyphenols present, such as betagarin or betavulgarin, affect the analysis and detection of betalains?
Response to 1). The presence of polyphenols in our analysis is limited since the extract of betalains is based on the use of MSPD with up to 50% methanol, thus leaving many metabolites that could have an impact on the analysis of betalains bound to the C18 beads.
Comment 2) Did you detect vulgaxanthins I and II?
Response to 2). Vulgaxanthin I or glutamine-bx were detected in our analysis, but not vulgaxanthin II or glutamic acid-bx.
Comment 3) Check the identification of gomphrenin II. The molecular formula does not match with the literature data (C33H32N2O15).
Response to 3). We were unable to detect gompherenin II, therefore it was not included in Table 1. Based on this comment, we double-checked Table S1 for the molecular formula of gomphrenin II and confirmed that it was displayed correctly.
Comment 4) Check the identification and naming of all glabranin-type betalains.
Response to 4). We double-checked Table 1 for glabranin-type betalains and confirmed that they are correctly displayed.
Comment 5) How can a polar (uncharged-type) betalain (71) elute later in reversed phase chromatography than hydrophobic-type betalains (72,73)?
Response to 5). We double-checked the Rt and Rt relative to betanin data and found an error in the annotated Rt and Rt relative to betanin for 73 and 74. The names 73 and 74 have been renamed to 75 and 76, respectively.
Comment 9: Correct Betaxhanitns to Betaxanthins in Table 1.
Response 9: Thank you so much for your comment. This error has been corrected.
Comment 10: Please, modify/clarify the text regarding collision energies used for CID in Figure 4 caption.
Response 10: Thank you so much for your comment. The CID value has been defined.
Comment 11: Please, modify Figures 5, 6 and 8 to be similar to Figure 4: add collision energies and structures of fragments on the top of the peaks in mass spectra.
Response 11: Thank you so much for your comment. The suggested changes to the figures have been made.
Comment 12: Please, add the chromatogram measured to Figure 7.
Response 12: Thank you so much for your comment. The suggested change has improved the figure.
Comment 13: Please, improve the layout of Table 1. Could it be horizontal to have more room for the title and compound names?
Response 13: Thank you so much for your comment. The suggested change has made Table1 more readable.
Comment 14: Please, add all parameters related to MassFrontier analysis. What does “The fragmentation parameters were established to be steric optimal, allowing resonance reactions for electron sharing, charge stabilization, and radical isomerization.” mean? Can you detect radical isomerization for betalains when CID at 50 eV is used?
Response 14: Thank you so much for your comment. All parameters about MassFrontier for in-silico fragmentation of betalains have already been deposited in the main text. Indeed, the software is displayed in boxes as check boxes. For instance, electron sharing is related to the way resonance occurs between bonds and atoms. More in depth, the vendor's user manual does not provide information on how the software was trained to perform this task. This is also happening to the other parameters.
In the software, there is no way to connect the amount of energy used to fragment betalains in mass spectrometry to the in-silico fragmentation of betalains. This may be possible with more research on fragmenting individual betalains in different fragmentation techniques (CID, HCD), different amounts of energy (10, 20, 25, 35), and different detectors used in mass spectrometry (Ion trap, orbitrap).
Comment 15: Conclusions: it is not now a database, but values in the supplementary material. Could you make it more accessible? For example, by adding the data to an already existing open access platform or to databases of Thermo Scientific (e.g. to be used via CompoundDiscoverer softaware)? You have made a huge and careful work but it is quite hard for the others to take an advantage of that.
Response 15: Thank you so much for your comment. We are exploring ways to make the database of betalains more meaningful in the future, and we anticipate the necessity of advanced bioinformatics. We think creating a database of betalains for those kinds of software (such as Compound Discover) dedicated to the analysis of metabolomics data would definitely be great. However, the mass spectrometry data generated in this work is limited to filling this type of database. Normally it requires the fragmentation of each analyte with different amounts of energy, for instance, CID with 10, 25, and 35, among other things.
Comment 16: In general, I do not feel that I’m qualified to access the quality of English but I would change some of the words used in the manuscript, like in the title: mass spectrometry fragmentation -> mass spectrometric fragmentation and fingerprint (the end result) -> fingerprinting (the technique).
Response 16: Thank you so much for your comment. The title has been changed based on the suggestions.
Sincerely,
Dr. Victor Aguilar-Hernández
Unidad de Biología Integrativa
Centro de Investigación Científica de Yucatán
Email: victor.aguilar@cicy.mx

Round 2
Reviewer 2 Report
Comments and Suggestions for Authors
Many thanks for your corrections and justifications! I have still a few minor comments:
line 150: delete m/z
Figure 7B: the chromatogram at 540 nm exhibits only 7 peaks with numbers but Table 1 contains 86 betalains. Does this mean significant coelution of betalains? Could the chromatography be improved? Can MS overcome this issue?
Regarding responses 6 and 14: thank you for the clarification. Just one small thing: the fragment ions in Table S1 were predicted by MassFrontier, which is now described in Materials & Methods as “Next, precursor masses [M+H]+ were created for in-silico fragmentation using MassFrontier 7.0 software (Thermo Scientific, USA).“ (lines 413-414). This could be clarified for the reader, i.e. the theoretical product ions were created….
Regarding response 7: I disagree that this is an innovative presentation. It is natural that the addition of something increases the molecular size and the loss of something decreases it. Perhaps I do not see the figure as you do and therefore, I honour your decision.
Regarding response 16: Please, reconsider the title (the repetition of fingerprinting/fingerprint).
Author Response
Reviewer
Dear Reviewer, we are grateful for your valuable feedback on our manuscript. The paper has been made better thanks to the comments, which we answered as best as we could.
Comment 1: line 150: delete m/z
Response 1: Thank you so much for your comment. We have made the correction.
Comment 2: Figure 7B: the chromatogram at 540 nm exhibits only 7 peaks with numbers but Table 1 contains 86 betalains. Does this mean significant coelution of betalains? Could the chromatography be improved? Can MS overcome this issue?
Response 2: Coelution of betalains is definitely present. Although MS analysis was helpful in detecting betalains of low abundance (looking at the base lane profiles of the monoisotopic mass ion and the isotopic mass ions detected in Orbitrap), it was not able to completely overcome the coelution issue. Further MS analysis may overcome this issue by using analytical methods that focus on enriching low-abundant betalains. The main reason why there are only 7 peaks instead of the expected 86 in Figure 7B is the low abundance of some betalains.
Comment 3: Regarding responses 6 and 14: thank you for the clarification. Just one small thing: the fragment ions in Table S1 were predicted by MassFrontier, which is now described in Materials & Methods as “Next, precursor masses [M+H]+ were created for in-silico fragmentation using MassFrontier 7.0 software (Thermo Scientific, USA).“ (lines 413-414). This could be clarified for the reader, i.e. the theoretical product ions were created….
Response 3: Thank you so much for your comment. We have made clarification in the main text, and it is now read as:
“Next, the structures were placed in the Structure Editor of MassFrontier 7.0 software (Thermo Scientific, USA). The charge specification was made to generate precursor masses [M+H]+ and fragment ions in-silico.”
Comment 4: Regarding response 7: I disagree that this is an innovative presentation. It is natural that the addition of something increases the molecular size and the loss of something decreases it. Perhaps I do not see the figure as you do and therefore, I honour your decision.
Response 4: Thank you so much for your comment. I would like to say that the figure has not been published anywhere, which is one of the reasons why we view it as innovative. I agree that it is obvious that there is an increase or decrease in molecular size in betalains when structural parts are added or removed from the framework, but it serves the scientist community who do not focus their research on betalains. We have decided to keep the figure in the main text.
Comment 5: Regarding response 16: Please, reconsider the title (the repetition of fingerprinting/fingerprint).
Response 5: Thank you so much for your comment. We removed the repetition of fingerprinting/fingerprint in the title, and it is now read as:
“In Silico Mass Spectrometric Fragmentation and LC-MS/MS Betalainic fingerprinting: Identification of Betalains in Red Pitaya”
Sincerely,
Dr. Victor Aguilar-Hernández
Unidad de Biología Integrativa
Centro de Investigación Científica de Yucatán
Email: victor.aguilar@cicy.mx
